# Does the COVID-19 Vaccination Rate Change According to the Education and Income: A Study on Vaccination Rates in Cities of Turkey between 2021-September and 2022-February

**DOI:** 10.3390/vaccines10111933

**Published:** 2022-11-15

**Authors:** Beyza Cengiz, Mustafa Ünal Sayılır, Nur Yıldız Zengin, Öykü Nehir Küçük, Abdullah Ruhi Soylu

**Affiliations:** 1Department of Pharmacology, Faculty of Medicine, University of Ankara, Ankara 06000, Turkey; 2Faculty of Medicine, University of Gazi, Ankara 06560, Turkey; 3Faculty of Medicine, University of Hacettepe, Ankara 06800, Turkey; 4Department of Biophysics, Faculty of Medicine, University of Hacettepe, Ankara 06800, Turkey

**Keywords:** COVID-19, vaccination, vaccine, income, education level, public health

## Abstract

Background: The increase in the coronavirus disease 2019 (COVID-19) vaccination rates in Turkey and in the world plays a key role in reducing the number of deaths and cases from COVID-19. Investigating the underlying causes of COVID-19 vaccine hesitations seems to be a guide in reducing the negative effects of the pandemic. Methods: We analyzed the correlations between double COVID-19 vaccination rates of all cities in Turkey between 1 September 2021 and 1 February 2022 and their per capita income values and their education level percentages. Results: Overall, there was a negative correlation between the vaccination rates of cities and the percentages of illiterate, literate without a diploma, and secondary school graduates for both genders. However, there was a positive correlation between city vaccination rates and the percentages of primary school and college graduates. City income values per capita values were positively correlated with double vaccination rates of cities. Conclusion: Our findings suggest that not only low levels of education, but also interruption of education at some point make a significant contribution to vaccination hesitancy and ultimately to vaccination levels. In order to end the pandemic and reach a sufficient percentage of vaccines, it may be necessary to address some special groups and raise awareness of these groups about vaccination.

## 1. Introduction

On 11 March 2020, WHO declared that COVID-19, the infectious disease caused by the SARS-CoV-2 virus, could qualify as a pandemic and called on all countries to activate and expand their emergency response mechanisms [1,2]. As of now, while COVID-19 has caused approximately 600 million cases and 7 million deaths in the world, it has also caused approximately 16 million cases and 100 thousand deaths in Turkey [3,4].

The role of the treatments used in this devastating picture created by COVID-19 has been limited [5]. Vaccination is critical in reducing the number of cases and deaths from COVID-19 and plays a key role in ending the pandemic [6]. If we take a look at the vaccine development process against the SARS-CoV-2 virus in the world; the Pfizer BioNTech (BNT162b2, Manhattan, NY, USA) COVID-19 vaccine became the first COVID-19 vaccine to be approved for emergency use by the World Health Organization as of 31 December 2020 [6]. Then, as of 12 January 2022, the emergency use of the Oxford/AstraZeneca (ChAdOx1-S (recombinant) vaccine) COVID-19 vaccine was approved by the World Health Organization [6]. Subsequently, the Janssen/Ad26.COV 2.S vaccine, produced by Johnson & Johnson, was approved on March 12, 2021. As of September 2022, there are 9 vaccines approved against COVID-19 [7]. According to the data announced by the World Health Organization, as of September 2022, the number of vaccines in clinical development is 171 and 45 of them are in phase 3 [8].

The inactivated COVID-19 vaccine Sinovac-CoronaVac was developed by Sinovac Life Science Company (Beijing, China). To evaluate the efficacy of the Sinovac-CoronaVac vaccine, participants were vaccinated in two doses 14 days apart in a randomized, double-blind, placebo-controlled phase 3 clinical trial among healthy healthcare professionals based in Brazil, conducted from 21 July to 16 December 2020. Efficacy against symptomatic COVID-19 for 14 days after the second vaccine dose was 50.7% (95% Confidence Interval: 36% to 62%) [9]. According to the results of a double-blind, randomized, placebo-controlled phase 3 trial to determine the efficacy and safety of the Sinovac-CoronaVac vaccine carried out in Turkey between 14 September 2020 and 5 January 2021, the effectiveness of the Sinovac-CoronaVac vaccine 14 days or longer after the second dose is 83.5% found [10]. Following these data, the Sinovac-CoronaVac vaccine was given “Emergency Approval” by the Ministry of Health of the Republic of Turkey, and then the COVID-19 vaccination process began on 14 January 2021 [11]. On 19 June 2021, the Ministry of Health announced that 5 million doses of the Sinovac-CoronaVac vaccine had arrived in Turkey [12]. On the other hand, the Sinovac-CoronaVac vaccine had been approved for emergency use by the world health organization on 1 June 2021 [7]. Priority groups were determined in the vaccination process in Turkey. In the first stage, the personnel working in the health institution and all pharmacy workers obtained the right to be vaccinated. Then, this right was granted to the elderly, the disabled, and citizens living and working in places such as protection houses. Afterwards, people over the age of 65 and priority sectors (education sector, food sector, transportation sector, security, ministry of justice, ministry of national defense, prisons, etc.) for the continuation of the service were determined and those working in these sectors were given the right to be vaccinated. Vaccination continued, expanding to lower ages [13]. After the Pfizer-BioNTech vaccine was approved for emergency use by WHO in December 2020, the Pfizer-BioNTech vaccine was offered to the citizens of the Republic of Turkey as the second vaccine alternative, after the Sinovac-CoronaVac vaccine, and the use of the Pfizer-BioNTech vaccine was started on 12 April 2021 [14]. The Pfizer-BioNTech vaccine is an mRNA vaccine against COVID-19 [15]. Between 27 July 2020 and 29 October 2020, in a placebo-controlled, observer-blinded, multinational study of participants aged 16 years and older with no prior evidence of SARS-CoV-2 infection, a 6-month follow-up was performed after two doses of the Pfizer-BioNTech vaccine given 21 days apart and the efficacy of the Pfizer-BioNTech vaccine against COVID-19 was 91% (95% Confidence Interval: 89% to 93%) [16]. While citizens living in the Republic of Turkey could only get the Sinovac-CoronaVac vaccine between 14 January 2021 and 12 April 2021, they were vaccinated with the Sinovac-CoronaVac or Pfizer-BioNTech vaccine as of 12 April 2021, according to their own preferences. On 30 April 2021, the Ministry of Health of the Republic of Turkey gave the emergency use approval for the use of Russia’s Sputnik V vaccine in Turkey after the Sinovac and BioNTech vaccines [17]. Although the first shipment of 400,000 doses of Sputnik V vaccine reached Turkey on 15 June 2021, the Turkish Medical Association announced that the vaccine could not be used because the second doses did not arrive [18]. Although it had been approved for emergency use as of September 2022, the Sputnik V vaccine is not in use in the Republic of Turkey. On 22 December 2021, Turkovac, which was developed in cooperation with Erciyes University and the Ministry of Health, Turkish Health Institutes, was approved for emergency use in Turkey and started to be implemented in the Republic of Turkey as of 30 December 2021 [19]. As a result, as of January 2022, the vaccines used against the SARS-CoV-2 virus in the Republic of Turkey are the Sinovac-CoronaVac, the Pfizer-BioNTech and the Turkovac.

On 25 June 2021, everyone aged 18 and over was given the right to be vaccinated. As of August 2022, approximately 85 percent of the population aged 18 and over who have received at least 2 doses of vaccination is approximately 45 percent of the population aged 18 and over who have received 3 doses of vaccination [20]. Although the vaccination percentage of the Turkish population is higher than the average vaccination percentage of the world population, the percentage of those who do not have the vaccination is also at a significant level [21].

Although there are many vaccines developed against COVID-19 that act with various mechanisms and there is no shortage of resources, the desired speed in vaccination programs has not been achieved in many countries [21]. Considering the role of time in the formation of SARS-CoV-2 variants, we think that the disruptions in the vaccine programs may have played a role in the emergence of new variants and the prolongation of the pandemic [22]. Taking into account the effect of the vaccine in preventing material and moral damages, it is critical to investigate the underlying causes of vaccine hesitations (not being sure about the vaccine). Researchers from many parts of the world have conducted studies to evaluate the attitudes of societies towards COVID-19 vaccines and to identify the factors that lead to vaccine hesitancy [23,24,25,26,27,28,29,30,31,32,33,34,35].

If we take a look at some of the elegant studies conducted in Europe, in a study based in Ireland and the United Kingdom, vaccine hesitancy was evident in 35% and 31% of adult participants, respectively, and the researchers concluded that those who were resistant to the COVID-19 vaccine in both populations were less likely to receive information about the pandemic from authoritative sources [23]. According to another study conducted in France, direct vaccine rejection and vaccine hesitancy were found to be significantly associated with lower education level, inadequate compliance with the vaccines recommended in the past, and not reporting the specified chronic conditions [24]. If we look at some of the comprehensive studies done in the Americas, in one study the intention to get vaccinated was 55% in January 2020 and around 70% in March 2020. When examining the population with high levels of vaccine hesitancy, it was found that young adults, non-Hispanic Black or other race adults, adults of low socioeconomic status, and adults living in the southeastern USA region [25]. In another US-based study involving participants aged 18–45 years, the outcomes of the population receiving the COVID-19 vaccine were associated with male gender, sexual minority status, higher education levels, and previous influenza vaccination [26]. A large-scale national study based in China, located in the Asian continent, examined COVID-19 vaccine hesitancy among the Chinese population. The overall prevalence of COVID-19 vaccine hesitancy in this study was found to be around 8%. It was concluded that women, those with a high level of education, married residents, those with high health status, those who had never smoked, and those with low belief in vaccination conspiracy were more willing to vaccinate than others [27]. In another study conducted to investigate hesitancy and resistance in the COVID-19 vaccine in the Australian continent, it was concluded that 7% of the participants had high hesitancy and 6% were resistant. Women, those living in disadvantaged areas, those with more populist views, and those with higher levels of religiosity have been shown to be more hesitant or resistant, while those with higher household incomes and more reliance on state or territory government or hospitals have been shown to be more likely to be vaccinated [28]. In a study on perspectives on COVID-19 vaccines in Jordan, a country located in the Middle East, a significant correlation was found between male gender, high income, master, doctorate, or undergraduate education level and willingness to have the vaccine [29]. In studies from all over the world examining the underlying causes of COVID-19 vaccine hesitancy, the main differences between the populations who have/intend to get vaccinated and the group who are hesitant about vaccination were their education levels, socioeconomic status, income level, the probability of getting information about the pandemic from authorized sources, and the health status of the participants.

Given that the Turkish government has stated that COVID-19 vaccine will be provided free of charge to the population, we questioned: is there a correlation between cities’ education levels, incomes and vaccination percentages? Since we used government data in this study, problems such as small sample size (several thousand), wrong population sampling (several thousand web/Facebook users, etc.) and dividing education levels into a small number of groups did not occur.

## 2. Materials and Methods

From 1 September 2021–1 February 2022, we examined correlations of double COVID-19 vaccination rates (DCVR) (over 18 years old) of all cities (81) in Turkey vs. their per capita income values and their education level percentages. Double COVID-19 vaccination means received two vaccine doses, BioNTech, Pfizer vaccine, or Sinovac COVID-19 vaccine. The COVID-19 vaccination data of more than 60 million Turks recorded by the Ministry of Health has been used in this study. Data on double COVID-19 vaccination rates of cities (between 0% and 100%) were obtained from the Turkish Ministry of Health database and were collected every two weeks between September 2021 and February 2022 [20]. Education data of the last four years [36] (2018 to 2021) were obtained from the Turkish Statistical Institute database. City education levels classified as illiterate, literate without a diploma, primary school (5 years), primary education (6–7 years), secondary school/intermediate graduate (8 years), high school graduate (12 years), college degree (universities and other higher educational institutions), master’s degree (including 5 or 6 years faculties) and doctoral degree. Population data at each education level for cities are normalized using the population of that city. The correlation between education data for each year and vaccination rates was calculated by Spearman’s rank correlation analysis. Separate analyzes were made for males, females, and for the total population. Per capita city income values (PCCI) of the last three years [37] (2018 to 2020) (USD) are obtained from the Turkish Statistical Institute database. Associations between COVID-19 vaccination rates and PCCI were examined through Spearman’s rank correlation analyze.

## 3. Results

As shown in Figure 1, for the education level vs. the DCVR correlations while the correlation values usually do not change with respect to the gender, vaccination rate measurement day and year of education data for each education level, correlations between DVCR, and education levels vary greatly from one education level to the next. As a result of all analyzes, a negative relationship was found between the vaccination rates of the cities and the percentages of illiterate, literate, and secondary school graduates for both genders. However, there was a positive correlation between city vaccination rates and the percentages of primary and university graduates for both genders.

For PCCI and DCVR correlations, as shown in Figure 2, the results found in analyzes with different vaccination rate measurement day and PCCI data calculated in different years were consistent. A positive correlation was found between PCCI and DCVR.

## 4. Discussion

Immunization, which is defined as a process that protects a person against a disease through vaccination, is one of the most effective ways to prevent infectious diseases and end pandemics [38]. Thanks to immunization, smallpox has been eradicated worldwide and polio has been eradicated in America [39,40]. The first vaccine produced in the world towards the end of the 1800s is the smallpox vaccine, and vaccine hesitancy/opposition started at the same time [41]. Therefore, vaccine hesitancy and refusal is as old as vaccination itself and is unlikely to disappear. In this context, we believe that the first step towards effective vaccination programs is to identify the causes and contexts that lead to vaccine hesitancy and rejection.

In this study, we tried to provide new information about COVID-19 vaccine levels, vaccine hesitations and underlying causes. Numerous studies have been published examining individuals’ attitudes towards COVID-19 vaccination and the reasons for exhibiting this attitude [23,24,25,26,27,28,29,30,31,32,33,34,35]. In a very large part of these studies, a limited number of people were reached online with the effect of the restrictions brought by the pandemic, and their attitudes towards vaccination were questioned through questionnaires [23,24,25,26,27,28,29,30,31,32,33,34,35]. We think that one of the reasons why our study has different results from other studies on this subject is that the data of the study is not the data obtained through a survey, but the data of the whole country and therefore it covers more people.

There are a lot of studies that contain data on the relationship between hesitations about vaccination and the educational status of individuals [23,24,25,26,27,28,29,30,31,32,33,34,35]. In most of these studies, including studies from Turkey [34], it was concluded that vaccination hesitations increased with the decrease in education level [24,25,26,29,30,32,33,34]. On the other hand, in these studies, the educational status of the participants was generally evaluated in 2, 3 or 4 categories [24,25,27,28,29,30,32,33,35]. In our study using government data, the educational status of the population over the age of 15 in the cities was evaluated in 9 different categories (illiterate, literate without a diploma, primary school, primary education, secondary school/intermediate graduate, high school graduate, college degree (universities and other higher educational institutions), master’s degree, and doctoral degree). In most of the previous studies, the categories of illiterate, primary school graduate, and secondary school graduate were handled under a single category and we think that the difference in our study is due to the more detailed categorization of educational status. Contrary to other studies, there was no positive correlation between education level and vaccination rates in our study. While there was a negative correlation between the vaccination rates of the cities and the percentages of illiterate people, literate without a diploma, and those who graduated from secondary school, a positive correlation was found between the percentages of university graduates and the percentages of primary school graduates.

Even though secondary school graduates have had a longer education period than primary school graduates, the difference in their attitudes towards vaccination reveals that there is no positive correlation between the increase in education period and vaccination in this study. This situation makes us think that the interruption of education at some point may be another important determinant in shaping one’s views on vaccination, along with the duration of education.

When we look at the relationship between per capita city income values and vaccination rates, which is another factor that we can access from the database of the Turkish Statistical Institute and which we think can guide people’s behavior regarding vaccination, we found a positive relationship between PCCI values and vaccination rates. Consistent with our study, in previous studies on this subject, when the income of the participants was questioned, vaccine hesitancy/resistance was higher in those with low income, and those with high income tended to be vaccinated [25,26,28,29,30,33]. Those with low incomes may be resistant to vaccination or have hesitations about vaccination because they could not access sufficient information about vaccination (vaccination options—vaccine efficacy, post-vaccination complications, etc.).

Only correlations (vaccination rate vs. education level, vaccination rate vs. PCCI) were calculated. As ‘correlation is not causation’, for more accurate analyses, while collecting vaccination data, other characteristics of the subjects such as detailed education levels should also be questioned and added to the research datasets. In addition, the quality and content of education, conservatism, religiosity, scientific literacy level etc. are potential variables and may have affected the vaccination level of the provinces. Unfortunately, we could not reach these data because the data for the provinces are limited. Another limitation of this study is that it does not include PCCI values for 2021.

## 5. Conclusions

A systematic review of how to approach vaccine hesitancy, based in the United States, concluded that the most effective interventions to reduce vaccine hesitancy are multi-component and dialogue-based interventions [42]. Therefore, we believe that defining the characteristics of the population to be addressed in interventions based on this dialogue in detail will increase the effectiveness of the strategy to be implemented. The current study shows that the education level of the populations has an impact on their attitudes towards the COVID-19 vaccine. Although awareness raising about vaccination is made for the whole society in order to increase the percentage of vaccination, the characteristics of some special groups should be considered and more efforts should be made to raise awareness of these groups.

Sudden jumps from negative to positive (or vice versa) correlation from one education group to another deserves more research. Investigating the effect of education level on COVID-19 vaccination in other countries may have important implications for increasing COVID-19 vaccination rates.

## Figures and Tables

**Figure 1 vaccines-10-01933-f001:**
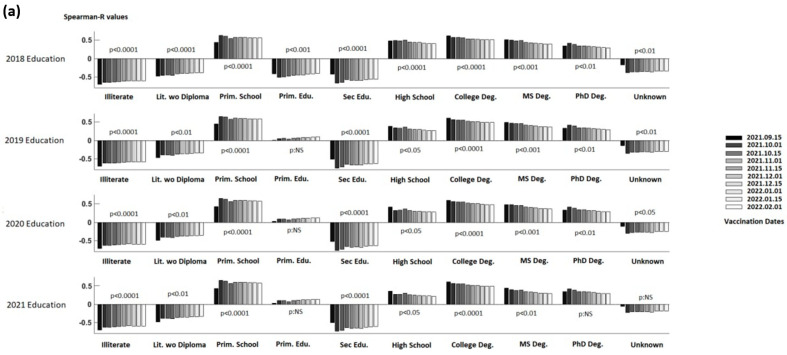
Spearman-R correlation coefficients (education percentage per city vs. double COVID-19 vaccination percentage in the city) for different educational levels (illiterate, literate without a diploma, primary school, primary education, secondary school/intermediate graduate, high school graduate, college degree, master’s degree, doctoral degree, and unknown) and years (2018–2021), between September 2021 and February 2022 COVID-19 vaccination measurement days ((**a**) Total, (**b**) Male, (**c**) Female).

**Figure 2 vaccines-10-01933-f002:**
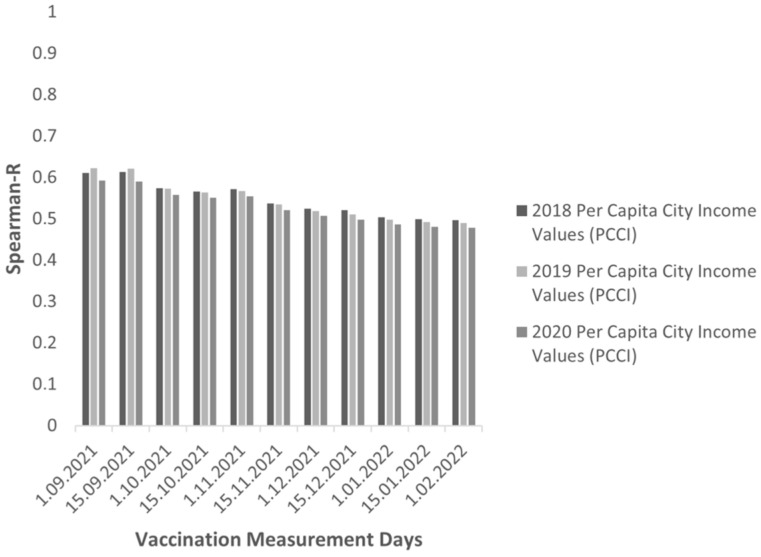
Spearman-R correlation coefficients (per capita city income values (2018–2020) vs. double COVID-19 vaccination percentage of cities (between 1 September 2021 and 1 February 2022 COVID-19 vaccination measurement days)) (All p values calculated for the city’s double COVID-19 vaccination percentage against per capita city income values are less than 0.0001).

## Data Availability

Republic of Turkey Ministry of Health COVID-19 Vaccine Information Platform https://covid19asi.saglik.gov.tr (accessed on 1 September 2022) website was accessed between 2021-September and 2022-February at 15-day intervals and city-based vaccination rates were obtained. The education level of the country over the age of 15 is obtained from the database of the Turkish Statistical Institute, https://data.tuik.gov.tr/Search/Search?text=e%C4%9Fitim&dil=1 (accessed on 1 September 2022)—”statistical tables” tab—“Education level completed by city (15+ years old)” tab (Data for the year 2018, 2019, 2020, 2021). Per capita city income values (PCCI) is obtained from the database of Turkish Statistical Institute, https://data.tuik.gov.tr/Bulten/Index?p=Il-Bazinda-Gayrisafi-Yurt-Ici-Hasila-2020-37188 (accessed on 1 September 2022)—“GDP per capita by province 2018-2020” tab (Data for the year 2018, 2019, 2020).

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
