# Peer review of "Does the COVID-19 Vaccination Rate Change According to the Education and Income: A Study on Vaccination Rates in Cities of Turkey between 2021-September and 2022-February"

_vaccines, 2022, doi:10.3390/vaccines10111933_

Round 1

Reviewer 1 Report

Referee report:

Your manuscript   definitely contains some useful information. However, before positive recommendation can be give, several important issues should be fixed

in order to better focus information flow and make important clarifications. 

Major comment 1: Please, be more precise in writing. This is a scientific journal!  

E.g. delete 1st sentence (page 1, lines 30-33), since it is confusing: in the reference [1.] link is not active and

for reference [2]: "can be characterized as a pandemic" does not meat standards for pandemic declarations, authors needs to check the standards of pandemic declarations. Also delete  sentence (page 3, lines 106-109)  "Disruptions...quite complex.": you did not make research on this.

Major comment 2:  positive correlation of PCCI values and the double vaccination is a fake correlation effect (you will get the similar with any positively increasing index approximately uniformly spread over the country. More detailed dependence analysis should be made, e.g. fitting a structural equation models or some asymmetric dependence measure. Also partial correlation analysis is needed. Checks of underlying assumptions (e.g. distributional assumptions)  of Spearman-R statistics will be also necessary

Major comment 3: Omit reference of [33] in Discussion. The methodology is not related to your work and it is confusing.

Major comment 4: Based on above mentioned changes, make a new rewrite of the article. Hesitation was/is completely justifiable/legitimate and you should position from independence angle to the topic, in order to eliminate biases. 

Author Response

Dear Reviewer, Thank you very much for your valuable comments. We have carefully reviewed your comments and have tried to answer your criticisms as best we can. We hope that we have been able to convey our answers to your criticisms in a sufficiently understandable language.We have attached the answers to your comments. Kind regards, Beyza Cengiz E-mail: [email protected]

Reviewer 2 Report

Congratulation to all authors for such an interesting and important research which is crucial for the policymakers and stakeholders to foster the vaccine uptakes in Turkey. Very well and clearly written in all the parts except in the results section. I would suggest making the results section, particularly Table 1 and Figure 1 which are not very reader-friendly. Mage this table and all Figure 1 in more readable form which is very hard to follow.

Paragraph 257 “Many factors were associated with vaccine resistance and hesitancy” does not look complete sentence so review this sentence and make it clear.

I have not seen the strengths and limitations of this manuscript so I would suggest adding the strengths and limitations of this study.

Author Response

Dear Reviewer,

Thank you very much for your congratulations and valuable comments. We have carefully studied your comments and have tried to respond to your criticisms as best we can. We hope that we were able to convey our answers to your criticisms in a sufficiently understandable language. We have attached the answers to your comments.

Kind regards,

Beyza Cengiz

Round 2

Reviewer 1 Report

Dear Authors, you did  a job in revision. Major comments 1 and 3 are satisfactorily answered.

Referee is sorry to hear that Professor Abdullah Ruhi Soylu, who assisted in the statistics section of the manuscript, is unable to support you due to an attack of Meniere's disease. I support to wait and  discuss my comment with the professor within additional  14-day period and to make an analysis.

Sincerely, Referee 

Author Response

Dear Reviewer,

Thank you very much for your understanding and valuable comments about the method used in statistics. We have carefully reviewed your comments.

As mentioned in our previous answer, most of the variables were not normally distributed (Anderson-Darling, D'Agostino&Pearson, Shapiro-Wilk, and Kolmogorov-Smirnov tests were used. For control, raw data were added to the system at the time of submission), so we used the non- parametric Spearman-Rank correlation. As we mentioned in the article (lines 277-285), we could not reach data on potential variables such as the quality and content of education, degree of religiosity, level of scientific literacy, how to access information about vaccination and source of information. etc.

Partial correlation analyzes were performed between the variables of education level and vaccination rates (by controlling the income variable). Since there are 3 different years, 9 different education levels and 11 vaccination measurement day, we added the results of 10 of all analyzes and we reached similar results in the others. If requested, we will send you the results of all analyzes (SPSS and Minitab were used.).

Additional analysis of income could not be performed because data on other variables that would affect the relationship between income and vaccination rate were not available.

Kind regards,

Dr. Beyza Cengiz
